# Extended fisheries recovery timelines in a changing environment

Gregory L. Britten[1,†], Michael Dowd[2], Lisa Kanary[3] & Boris Worm[1]

Rebuilding depleted fish stocks is an international policy goal and a 2020 Aichi target under the Convention on Biological Diversity. However, stock productivity may shift with future climate change, with unknown consequences for sustainable harvesting, biomass targets and recovery timelines. Here we develop a stochastic modelling framework to characterize variability in the intrinsic productivity parameter ($r$) and carrying capacity ($K$) for 276 global fish stocks worldwide. We use models of dynamic stock productivity fitted via Bayesian inference to forecast rebuilding timelines for depleted stocks. In scenarios without fishing, recovery probabilities are reduced by 19%, on average, relative to models assuming static productivity. Fishing at 90% of the maximum sustainable rate depresses recovery probabilities by 42%, on average, relative to static models. This work reveals how a changing environmental context can delay the rebuilding of depleted fish stocks, and provides a framework to account for the potential impacts of environmental change on the productivity of wildlife populations more broadly.

[1] Department of Biology, Dalhousie University, Halifax, Nova Scotia, Canada B3H 4R2. [2] Department of Mathematics and Statistics, Dalhousie University, Halifax, Nova Scotia, Canada B3H 4R2. [3] Yukon Research Centre, Yukon College, Whitehorse, Yukon, Canada Y1A5K4. † Present address: Department of Earth System Science, University of California, Irvine, California 92697, USA. Correspondence and requests for materials should be addressed to G.L.B. (email: gbritten@uci.edu).

With widespread recognition of the economic and ecological risks caused by progressive depletion of global fisheries[1–4], scientists and policy makers have shifted their focus to the rebuilding of depleted stocks[5–7]. Rebuilding initiatives have received major international support, beginning with the Magnuson–Stevens Fishery Conservation and Management Act in the United States[7], followed by major fisheries reform in Europe[8], and the declaration of an explicit international Aichi target for 2020 under The Convention on Biological Diversity[9]. The unifying target is to rebuild stocks to the biomass that produces the theoretical maximum sustainable yield ($B_{MSY}$, see Methods). We hereafter use the term 'recovery' to refer to the event where a depleted stock grows to exceed the $B_{MSY}$ target.

Explicit recovery targets and timelines require the ability to predict contemporary and future stock productivity. However, there is an increasing recognition that ongoing environmental change is already having an impact on fish population dynamics[10–14] and that recovery targets and timelines may in fact be moving targets. This challenges the commonly held assumption that populations are in a long-term steady state[15], wherein the production of biomass varies interannually but is governed by biological parameters that are stationary over time. While this assumption may have been questionable *a priori*, recent empirical analyses of global fisheries time series have suggested that significant non-stationary behaviour is occurring in the majority of global fish stocks in response to environmental change[13], while persistent regime-like behaviour[14] can cause unexpected collapse of otherwise tightly controlled populations[16]. Despite these observations, it remains unclear how non-stationary productivity dynamics have altered recovery timelines for global fish populations currently below target biomass levels. A previous meta-analysis of recovery timelines assumed static productivity[7], and therefore does not account for environmentally and biologically driven changes occurring today[13,14,16].

Here we analyse global fisheries time series to investigate non-stationary productivity in exploited fish populations and then evaluate the consequences for recovery of stocks currently depleted below target biomass. We utilize a Bayesian hierarchical modelling framework to relax the assumption of stationary (fixed) biological parameters, and allow those parameters to shift over time. We apply this framework to a simple and widely applicable production model to draw inferences across a variety of stocks and regions using a global database of population time series. We find that most populations are indeed non-stationary and highlight how this variation, if unrecognized, can cause 'silent' over- or underfishing. We further demonstrate how recovery targets and timelines must be adapted to account for the effects of environmental change today and into the future.

## Results

**Modelling non-stationary productivity.** We develop a non-stationary analysis of the intrinsic productivity parameter $r$ using fisheries time series for 276 stocks worldwide from the RAM Legacy Stock Assessment Database[17] (note that a 'stock' is defined here as population unit under management and may represent one or more populations within a metapopulation). We base the analysis on the foundational Graham–Schaefer surplus production model

$$B_{t+1} = B_t + rB_t\left(1 - \frac{B_t}{K}\right) - C_t + e_t^B, \qquad (1)$$

where $B$ denotes stock biomass, $C$ is the harvested biomass (the catch), with subscripts $t$ denoting years, $r$ is the intrinsic

productivity of the stock, $K$ is the carrying capacity and $e_t^B$ is the annual biomass deviation due to unresolved processes and measurement error, statistically described by a Gaussian distribution with a mean of zero and variance $\sigma_B^2$, or $e_t^B \sim N(0, \sigma_B^2)$. Labelled 'the most fundamental of all ecological parameters'[18], the magnitude of $r$ controls the growth trajectory of a population, maximum sustainable fishing mortality and the timeline to recovery. The carrying capacity $K$, in contrast, sets the long-term maximum attainable biomass and related reference targets such as $B_{MSY}$.

To investigate non-stationary dynamics in stock productivity and recovery in global stocks, we first extracted the annual biomass deviations from the static Graham–Schaefer model $(e_t^B)$ and fit a first-order autoregressive time series model of the form $e_t^B = \alpha e_t^B + \delta + \varepsilon_t$, where $\varepsilon_t \sim N(0, \sigma_\varepsilon^2)$. Under noninformative priors, we fit the model to all individual stocks and then classify a stock as non-stationary if the 95% posterior credible interval for the autoregressive coefficient α does not contain zero. This biological definition of stationarity differs from statistical stationarity in the following way: a statistical process is stationary if its probability distribution is invariant in time, that is, $e_t^B = \alpha e_t^B + \delta + \varepsilon_t$ is stationary if $|\alpha| < 1$ and $\delta = 0$, yielding $E[e_t^B] = 0$ and $Var[e_t^B] = \frac{\sigma_\varepsilon^2}{1 - a^2}$ for all $t$, with an autocovariance function that depends only on lag. Here we define biological nonstationarity as a stochastic process with temporal memory (autocorrelation) such that annual biomass deviations contain persistent regime-like behaviour, which is well-modelled by an autoregressive process with positive α (ref. 19).

For stocks characterized as biologically non-stationary, we capture variation in the autocorrelated deviations of annual biomass via a time-varying intrinsic productivity parameter modelled as a stochastic random walk with drift

$$r_{t+1} = r_t + \delta + e_t^r \qquad (2)$$

where $r$ is the intrinsic productivity state in years $t$ and $t+1$, $\delta$ is the drift and $e_t^r$ is the stochastic annual productivity forcing with distribution $e_t^r \sim N(0, \sigma_r^2)$ that governs the variability in productivity from year to year. We formulate the non-stationary Graham–Schaefer as a hierarchical Bayesian model[20,21] to reconstruct characteristic productivity variation for all depleted stocks and estimate the key parameter $\sigma_r^2$ for each stock, which quantitatively characterizes the response to biologically and environmentally driven productivity forcing. We fit the Bayesian model via Markov Chain Monte Carlo (MCMC) using the software Stan[22]. Mathematical details of the model are given in the Methods.

To forecast the recovery time for depleted stocks under the assumption of non-stationary productivity, we computed posterior predictive ensemble forecasts for future biomass by simulating random walk trajectories for the intrinsic productivity $r$ starting from the most recently estimated productivity state. In this way, the magnitude of historical variation in productivity $(\sigma_r^2)$ and the current productivity state are accounted for, while the direction of productivity change remains unknown. We assessed stock recovery 10 years into the future following the most recent observation since this is the default timescale for recovery planning in the US Magnuson–Stevens Fishery Conservation and Management Act[7]. We note that general conclusions apply equally to other recovery timescales.

To investigate the effects of time-varying carrying capacity $K_t$, we also computed ensemble forecasts using the non-stationary Graham–Schaefer model where both $r_t$ and $K_t$ vary simultaneously as stochastic random walks. We performed forecast simulations where stochastic variation in $K_t$ was assumed to be 5, 10 and 20% of the mean posterior $K$ for each individual stock

$(\sigma_K^2 = 0.05\bar{K}, 0.10\bar{K}, 0.20\bar{K})$, while $r_t$ varied according to the estimates of $\sigma_r^2$ from the Bayesian inference. On the basis of initial simulation studies, we found that known (simulated) changes in the underlying carrying capacity and its variance $\sigma_K^2$ were more difficult to re-estimate relative to changes in $r_t$ (Supplementary Fig. 1) because of the weak dynamical connection between biomass and $K_t$ when biomass is low. In contrast, $r_t$ is directly proportional to the stock rate of change across a wide range of biomass levels, yielding more reliable estimation when averaging over possible biomass states. For these reasons we focused the formal time series estimation on $r_t$ and present models of non-stationary $K_t$ in sections of the paper on biomass forecasting and recovery timelines.

**Patterns in stock productivity and their consequences**. The analysis and classification of annual biomass deviations with respect to the static Graham–Schaefer fit revealed that 68% of all stocks exhibited significant non-stationary behaviour (Fig. 1a), suggesting persistent regimes in stock productivity for global stocks. Of stocks that are depleted ($B < B_{MSY}$), 61% were classified as non-stationary (Fig. 1b). The mean autocorrelation parameter (indicating the persistence of productivity) across all stocks was $\bar{\alpha} = 0.45$ (95% credible interval: 0.39, 0.49), and for depleted stocks $\bar{\alpha} = 0.41$ (0.36, 0.47). While many individual stocks exhibited a long-term productivity trend ($\delta$), the overall mean

was not appreciably different from zero, indicating no directed long-term change in globally averaged productivity over time (Fig. 1c,d).

Examples of two iconic fish stocks highlight the practical consequences of productivity variation in a changing environment (Fig. 2). The Gulf of St Lawrence Atlantic cod (*Gadus morhua*) stock collapsed and was placed under moratorium in the early 1990s after realized surplus production had gone negative, but catches remained high[23] (Fig. 2a,c). Similarly, bluefin tuna was proposed for a CITES trade ban in 2010, following a period of low realized productivity and rampant overfishing in the early 2000s (ref. 24; Fig. 2b,d). Inferred trends in surplus production and sustainable yield (dashed lines, Fig. 2) were observed as long-term directional changes in the case of cod (Fig. 2a,c) or as oscillating regimes in tuna (Fig. 2b,d). In both cases, the biomass available for harvest was at times systematically over- or underestimated in a static productivity framework and likely led to periods of unrecognized, or 'silent', over- and underfishing (red and grey shading in Fig. 2, respectively). Fixed harvest strategies effectively reproduce the long-term historical average productivity from non-stationary models (equal red and grey areas, Fig. 2) and, therefore, poorly capture contemporary productivity over much of the observed period. These examples illustrate how changes in the underlying stock productivity are ignored by static models but can be tracked within a non-stationary framework.

For depleted populations, systematic productivity variation suggests that recovery should be forecast from recent observations that best characterize the current productivity state and its variability, rather than from observations back in time. Examples of non-stationary biomass forecasting are provided in Fig. 3. Historical variation in $r_t$ (characterized by $\sigma_r^2$) is estimated from the observed time series such that the interannual changes in biomass are accurately predicted within measurement uncertainty (Fig. 3a,b). Ensemble forecasts based on realizations of the non-stationary productivity model are then propagated into the future with associated biomass predictions (Fig. 3c,d). Note that the positive contemporary productivity regimes for Gulf of Alaska Pacific cod results in upward trending biomass forecasts, while in contrast, biomass trajectories are trending downwards for North Sea herring, which occupies a negative productivity regime at the end of the observed period (Fig. 3c,d). The recovery probability for a stock after a period into the future is taken from the relative overlap in the histograms with respect to the posterior distributions of the forecasted biomass and the estimated rebuilding target (Fig. 3e,f).

Under the assumption of no fishing ($F = 0$), posterior ensemble forecasts estimate that 57% of depleted stocks are predicted to recover within a 10-year time window (defined as having probability of recovery greater than half; Fig. 4a); however, appreciable uncertainty exists in the ensemble and some proportion of all stocks' trajectories do not recover because of a nonzero probability of low-productivity regimes in the future (Fig. 3). This contrasts with results from traditional static productivity model when the intrinsic productivity ($r$) is assumed to be stationary (for example, ref. 7). In this case, the mean recovery time is $\sim$3 years and recovery time probability is deterministic—either a stock recovers or it does not. The number of stocks recovering in the deterministic case is 78% (Fig. 4a). When we investigated the effect of non-stationary carrying capacity ($K_t$), we generally found that variation in $K_t$ had the effect of reducing the recovery time probabilities for all stocks (Supplementary Fig. 2) relative to static $K$. This effect was positively related to the magnitude of $\sigma_K^2$ (Supplementary Fig. 1c–h). Recovery time probabilities were reduced by 3%, 5% and 11%, on average, for $\sigma_K^2 = 0.05\bar{K}, 0.10\bar{K}, 0.20\bar{K}$, respectively.

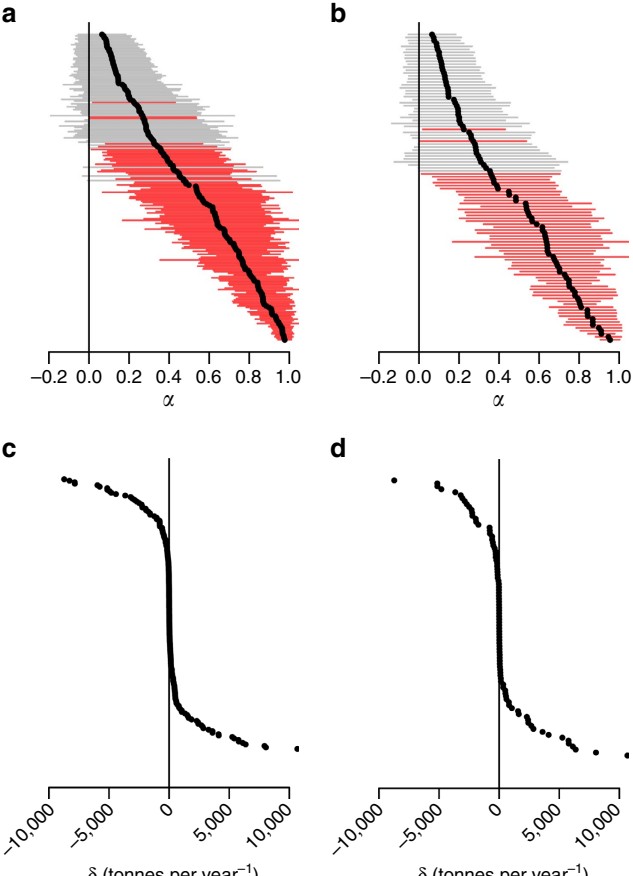

**Figure 1 | Meta-analysis of static productivity residuals across fish stocks.** Shown are the estimated autoregressive parameter $\alpha$ (**a,b**) and stochastic trend parameter $\delta$ (**c,d**) fitted to the biomass residuals of the static Graham–Schaefer model for all 276 stocks (**a,c**) and those stocks depleted below $B_{MSY}$ (**b,d**). The horizontal lines in **a,b** represent the 95% credible intervals, with red lines representing cases that do not overlap zero. Plotting of credible intervals is suppressed for **c,d** for presentation purposes.

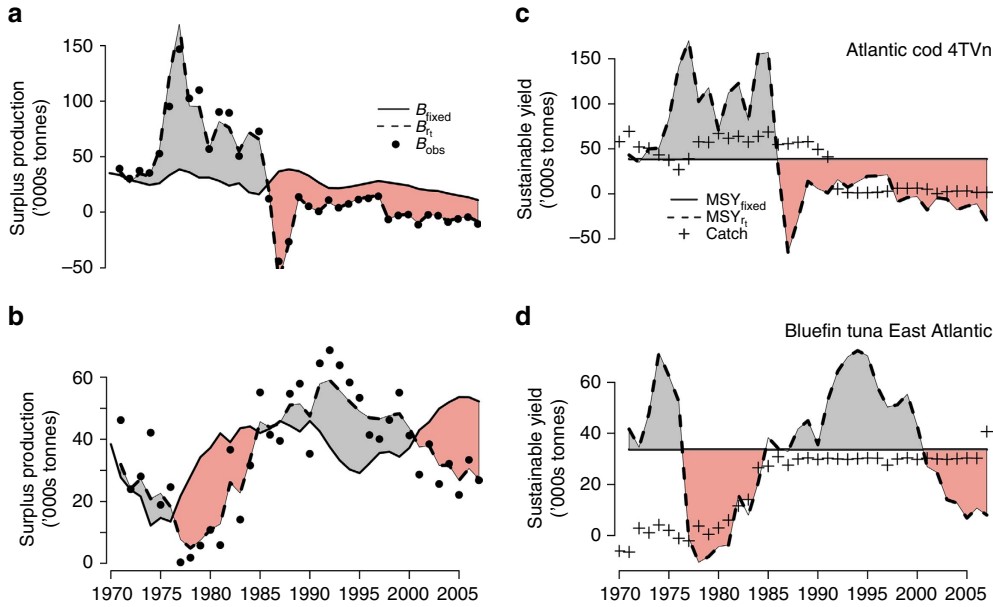

**Figure 2 | Examples of non-stationary productivity.** Two example stocks are shown: Atlantic cod from the Southern Gulf of St. Lawrence in Canada, fisheries region 4TVn (**a,c**) and bluefin tuna from the East Atlantic (**b,d**). (**a,b**) The annual surplus production for each stock (circles are observed values, $B_{obs}$, solid line is biomass-predicted from the stationary model with fixed $r$, denoted $B_{fixed}$, and the dashed line is the biomass-predicted from the non-stationary model, $B_{rt}$). (**c,d**) The theoretical maximum sustainable yield (crosses are recorded catches). Grey shading indicates when productivity is higher than would be predicted based on a static productivity model (potential underfishing) and red shading indicates lower-than-expected productivity that would promote overfishing.

When fishing a depleted stock at 90% of the maximum sustainable rate ($F = 0.9F_{MSY}$), only 18% of stocks are predicted to recover within 10 years in our analysis (Fig. 4b); this contrasts with 66% predicted to recover over the same timeframe under assumed stationary (fixed) productivity (note that the effect of $K_t$ was similar under both fishing scenarios). Interestingly, the mean biomass trajectories were not qualitatively different if $F_{MSY}$ is fixed at a static mean value ($F = 0.9F_{MSY}$) or allowed to vary according to $F = \frac{0.9r_{t-1}}{2}$, which represents a crude adaptive fishing strategy that is updated according to the most recent estimate of time-varying productivity. The variance of biomass was, however, generally higher under the fixed fishing strategy with potentially important consequences for fisheries management. We emphasize that modelling optimal harvest strategies under non-stationary population dynamics should be a focus of future fisheries management research.

## Discussion

Our analyses revealed previously undescribed biological variability in the productivity of global marine fish populations, which acts to increase the uncertainty in recovery timelines for currently depleted fisheries and can lead to systematic over- or underfishing in otherwise well-managed stocks. While time-varying methods are already being adopted in the analysis of individual stocks[25,26], this analysis is the first to quantify the extent of non-stationary production across all assessed stocks globally and chart the consequences for current rebuilding targets for those depleted. Accounting for non-stationary behaviour reveals that only one in five depleted stocks are predicted to recover over the next decade when being fished at a reasonable rate of $0.9F_{MSY}$. This provides important context for current rebuilding targets under the Convention on Biodiversity and the United Nations Sustainable Development Goals[9].

Our results are supported by a recent global analysis of biomass residuals[14] that found frequent statistical regimes in the residuals

of static productivity models with no overall directional trend, while there is also broad evidence of environmentally driven declines in recruitment capacity across global stocks[13]. Because recruitment, individual growth and natural mortality combine to determine total productivity, non-stationary trends in growth or natural mortality may compensate for declining recruitment, for example, reduced predation via predator release[27-29], which may explain the lack of long-term global productivity trends. Changes in total productivity may also be lagged relative to changes in recruitment, and therefore may not be felt until weak year classes and their progeny constitute a larger fraction of fishable biomass. These basic questions in fishery ecology highlight the need for more mechanistic studies into the individual drivers of observed non-stationary productivity patterns[16].

At the level of individual stock management, static models bias management towards a theoretical average productivity state that can lead to inadvertent mismanagement (Fig. 2) and unrealistic recovery timelines (Figs 3 and 4). The consequences of ignoring such changes may vary in severity, from suboptimal harvesting of the resource to potential collapse of an otherwise well-managed stock, as recently shown for Gulf of Maine cod[16]. Our analysis suggests that harvesters and managers must recognize that stocks may not recover to their previous state under non-stationary environmental conditions. More sophisticated models of productivity variation may be used to provide detailed stock-specific prediction, but this will only reinforce the need to rethink management expectations and build contingencies to account for progressive changes in stock productivity. Modern stock assessment methodology is beginning to meet the need to model changes in underlying parameters, most notably in the Stock Synthesis software package[26], which allows options for non-stationary parameters related to catchability, natural mortality and stock recruitment. Several authors have also advocated for a hierarchical time series approach like that developed here to detect biological shifts in fishery parameters and provide stock-specific management advice[26,30-32]. We heeded these suggestions

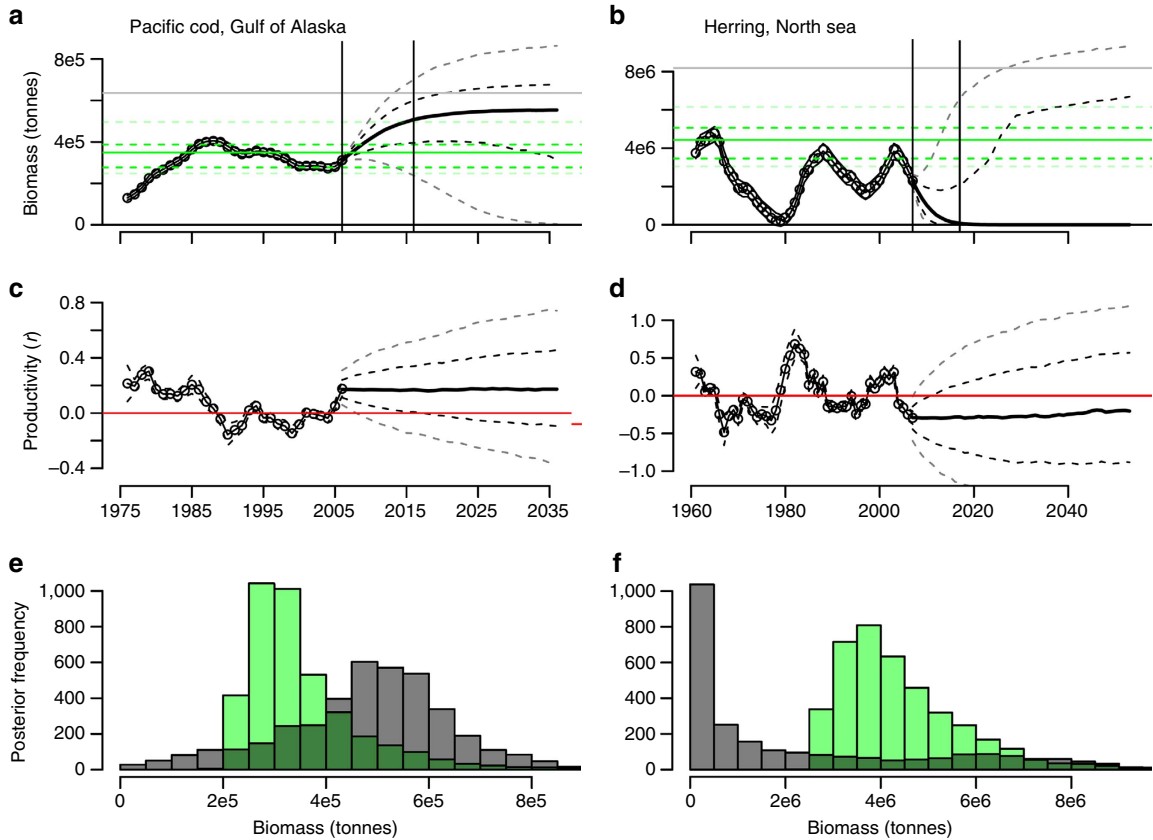

**Figure 3 | Examples of recovery timeline forecasting.** (**a,b**) Two examples of non-stationary biomass time series to which the non-stationary Graham–Schaefer model was fit. The solid grey lines indicate the mean posterior carrying capacity $K$; solid green lines give the mean posterior $B_{MSY}$ (with 50 and 90% intervals as dotted green lines). Vertical black lines differentiate the observed period and 10 years after. The mean biomass trajectories beyond the observed period are given as solid black lines with 50% and 90% intervals as dashed black and grey lines, respectively. (**c,d**) The fitted and forecasted productivity trajectories that drive the biomass predictions (red lines indicate lines of zero net growth). The posterior predictive distribution for recovery after 10 years is then taken from the sample histograms in **e,f**, where the biomass forecasts are given in grey and the $B_{MSY}$ posterior estimates given in green.

but took a multispecies meta-analytic approach to quantify the extent of biological change in global production time series and the consequences for recovery prospects in depleted stocks worldwide. The observed prevalence of non-stationary behaviour in depleted stocks strengthens the need for adaptive fishery management approaches in order to ensure the rebuilding of global fisheries in a changing environment.

Our hierarchical Bayesian approach provides a flexible framework for such adaptive management, based on established statistical theory used in analogous non-stationary systems such as weather prediction and real-time target-tracking[20,33]. A key attribute is the seamless ability to incorporate new observations as they become available to track a time-varying distribution for the underlying model state. While our results support the ongoing extension of operational stock assessment software to model non-stationary parameters for specific stocks[26], we note that the hierarchical Bayesian approach can readily incorporate non-stationary parameters into any quantitative wildlife management model to capture dynamic changes in key vital rates. To do so, the traditional model is restructured hierarchically with an additional set of ecologically interpretable variance parameters describing the range of potential for change in biological rates over time. Bayes' theorem is then used to directly update the time-varying distribution as observations become available[20,33]. The approach applies across a variety of contexts, including parameter reconstruction from historical observations, real-time management intervention, and future prediction. We view it as

a major advantage that the basic machinery of the traditional model remains intact and managers do not require complex ecosystem models or new data streams to incorporate environmental change into adaptive management advice. Importantly, however, such a dynamic approach to population modelling may require a parallel shift in management procedures to accommodate potentially rapid changes in fishing pressure and effort re-allocation within the fishery. For example, equilibrium-based reference points, such as long-term maximum sustainable yield, will inevitably be less stable in a non-stationary context and their use may lead to inefficient management and poor socioeconomic outcomes. Time-varying management reference points derived from the current productivity state[32] may therefore provide a more suitable approach to manage stocks in a non-stationary context.

As opposed to total productivity ($r$), which primarily governs the interannual variability of stock biomass, variation in carrying capacity ($K$) shifts the target to which stock biomass must recover. While processes altering stock carrying capacity are undoubtedly taking place today (for example, habitat contraction[34,35]), they appear to be more difficult to detect from biomass time series alone (Supplementary Fig. 1). From a policy perspective, changes in biomass targets due to $K_t$ are problematic since changes in $K_t$ alter the target to which managers are trying to rebuild. For both statistical and ecological reasons outlined above, auxiliary knowledge of the processes affecting $K_t$ (for example, spatial distribution) appears particularly

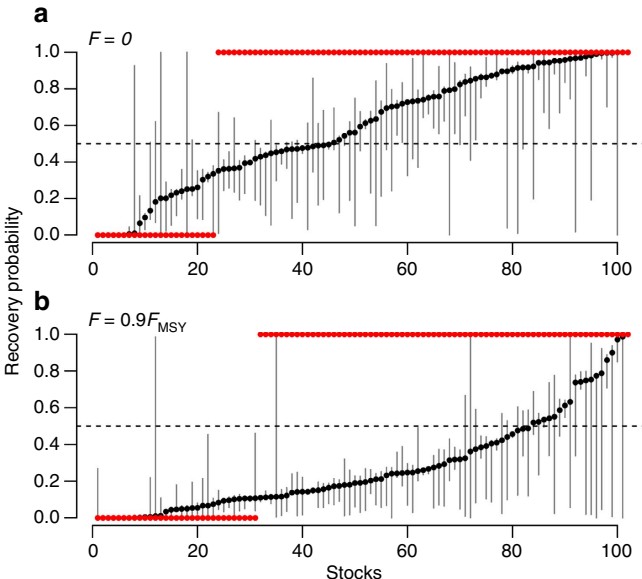

**Figure 4 | Consequences of non-stationary productivity for recovery of currently depleted stocks.** Shown are recovery probabilities over a 10-year timeline under no fishing ($F = 0$; **a**) and under fishing at 90% of maximum sustainable yield ($F = 0.9F_{MSY}$; **b**). Black circles indicate probability point estimates; vertical lines display uncertainty, where the lower (upper) limit is assessed as the proportion of trajectories exceeding the 10% (90%) posterior $B_{MSY}$ quantile. For comparison, red circles give the predicted recovery probability for each stock based on the deterministic Graham–Schaefer theory where the intrinsic productivity is fixed at its mean historical value. In the deterministic case, a stock recovers with probability 1 or 0.

important for constraining interannual variation in this parameter. This again suggests that adaptive fishery reference points based on the time-varying productivity state[32] should be favoured relative to long-term equilibrium reference points (for example, $B_{MSY}$). While the general result of increased recovery uncertainty under non-stationary productivity is largely independent of variation in $K$, we emphasize that developing a quantitative understanding of changes in biomass targets is a critical area for future work.

Internationally, the Magnuson–Stevens Fishery Conservation and Management Act of the United States is unique in legally mandating that depleted stocks must be rebuilt within a set timeframe of being declared overfished[7], thus imposing legal ramifications to the estimation of quantitative recovery timelines. Revealing non-stationary dynamics across the majority of depleted stocks globally, our results highlight pervasive biological variability in fishery time series that expands the uncertainty window for global fishery management (Fig. 4). Moderate fishing mortality was shown to slow and often halt recovery in these forecasts. The management implications for failing to meet set recovery timelines likely vary from fishery to fishery, but may include over- or under-capacity, inefficient effort re-allocation, and unintended overfishing. While the calculations presented here are necessarily simplified to compare productivity across disparate stocks, regions and data availabilities, they highlight the need for more precautionary rebuilding plans that better account for environmental change affecting stock productivity.

In conclusion, ongoing climate change challenges traditional assumptions about the future stability and predictability of fisheries production from the oceans. Our global meta-analysis of non-stationary production dynamics in depleted fish stocks tracks

patterns of productivity variation in global fishery time series and provides a framework to incorporate biological uncertainty into stock-rebuilding plans and wildlife management more broadly. As ocean conditions continue to change, our results help provide an empirical basis for adaptive fishery management that should aid in sustainably harvesting fish populations, rebuilding depleted stocks, and meeting international biodiversity targets.

## Methods

**Data.** The time series analysed in this study were extracted from the RAM Legacy Stock Assessment Database[17] (www.ramlegacy.org), which is a quality-controlled compilation of global fishery data. We extracted interannual time series for 276 stocks for which there was a direct estimate of total stock biomass ($B_t$) and catch ($C_t$). Some stock assessment methodologies produce 'retrospective analyses' where estimates of stock biomass are provided prior to the period when biomass surveys were performed. We excluded retrospective periods from 31 individual time series on a stock by stock via visual inspection (retrospective periods are identified as being characteristically 'smooth'). Individual time series length varied with a mean start year of 1971 and a mean end year of 2010. An overview of all stocks used in the analysis is given in Supplementary Table 1 along with its unique identifier in the RAM database and the most recent estimate of stock status $B_0:B_{MSY}$.

**Static Graham–Schaefer analysis.** To initially investigate and classify stocks based on the stationarity of biomass residuals, we fit the static Graham–Schaefer by maximizing the log posterior under noninformative priors for $r$ and $K$, assuming Gaussian distributed model errors for stock biomass. The residuals of the static fit with respect to the maximum *a posteriori* parameter estimates were then extracted and analysed with the autoregressive model described in the main text.

**Bayesian formulation of the non-stationary Graham–Schaefer model.** We extended the static Graham–Schaefer surplus production framework by formulating a time-varying intrinsic productivity state as a stochastic random walk with drift

$$B_{t+1} = B_t + r_t B_t \left(1 - \frac{B_t}{K}\right) - C_t + e_t^B, \qquad (3)$$

$$r_t = r_{t-1} + \delta + e_t^r \qquad (4)$$

where $B$ is the biomass in year $t$, $r_t$ is the non-stationary productivity with $e_t^r \sim N(0, \sigma_r^2)$ and stochastic trend $\delta$, $K$ is the carrying capacity, $C_t$ is the observed catch. Assigning probability distributions to the unknown quantities, the non-stationary Graham–Schaefer model can be written as the following hierarchical Bayesian state space model

$$p(B_{t+1}|B_t, r_t, K, C_t) \sim N\left(B_t + r_t B_t \left(1 - \frac{B_t}{K}\right) - C_t, \sigma_B^2\right) \qquad (5)$$

$$p(r_t|r_{t-1}) \sim N\left(r_{t-1} + \delta, \sigma_r^2\right) \qquad (6)$$

$$p(K) \sim N_{\text{trunc}}\left(K_0, K_0^2\right) \qquad (7)$$

$$p(\sigma_B^2) \sim \text{Uniform}(0, B) \qquad (8)$$

$$p(\sigma_r^2) \sim \text{Uniform}(0, \bar{r}_{obs}) \qquad (9)$$

where $K_0 = \max(B_t + C_t)$. The notation for the uniform densities represents equal *a priori* probability over the continuous interval $(a, b)$. Note that $N_{\text{trunc}}$ specifies a truncated Gaussian where the random variable is numerically constrained to be non-negative. Also note that the weakly informative empirical priors are set to aid MCMC convergence as typical biomass varies by orders of magnitude across stocks, making it difficult to specify noninformative priors across all time series.

**Recovery times.** Static and deterministic surplus production theory derives optimal harvesting by postulating a steady-state balance between harvest and production ($B_{t+1} - B_t = 0$) where the catch is equal to production $C_t = r B_t \left(1 - \frac{B_t}{K}\right)$ and yields the familiar expression where a biomass of half the carrying capacity produces maximum yield, $B_{MSY} = \frac{1}{2}K$. For a stock that is depleted below $B_{MSY}$, the deterministic stock recovery time is found by solving the initial value problem for the static Graham–Schaefer model with the initial condition $B = B_0$, where $B_0 < B_{MSY}$, and setting the solution equal to $B_{MSY}$, yielding an expression for the number of years need to recover[7]

$$t_{B > B_{MSY}} = \ln\left(\frac{2(1-\theta)(B_{MSY}/B_0) - 1}{1 - 2\theta}\right)\frac{1}{r - F} \qquad (10)$$

where $\theta$ is the ratio of fishing mortality to productivity $\frac{F}{r}$. Assuming that $r$ and $K$ are known and fixed, $t_{B > B_{MSY}}$ predicts the expected number of years as a deterministic function of current biomass $B_0$, $B_{MSY}$ and the level of fishing $F$ (Fig. 4).

Under non-stationary parameters, $t_{B>B_{MSY}}$ is time-dependent and its probability distribution is strongly positively skewed, which provides a poor metric to characterize expected stock recovery. Under the more realistic assumption in which $r_t$, $K$ and the system variance parameters are not known and must be estimated, the posterior predictive distribution for the recovery time requires sampling over the posterior-weighted predictive distribution

$$p\left(B_{\tilde{T}} > B_{MSY} | B_{obs}\right) \propto$$
$$\int p\left(B_{\tilde{T}} > B_{MSY} | B_{1:\tilde{T}}, r_{1:\tilde{T}}, \theta, B_{obs}\right) p\left(B_{1:\tilde{T}}, r_{1:\tilde{T}}, \theta | B_{obs}\right) p\left(r_{1:\tilde{T}}, B_{1:\tilde{T}}, \theta\right) \mathrm{d}r_{1:\tilde{T}} \mathrm{d}B_{1:\tilde{T}} \mathrm{d}\theta,$$

(11)

where $\tilde{T}$ is the is the recovery timescale and $\theta = \left\{K, \sigma_r^2, \sigma_B^2\right\}$ are the static parameters. Within the MCMC framework, we computed the posterior predictive density by Monte Carlo forward propagation of the random walk productivity process to generate sample realizations for 10 additional years beyond the scope of the observed time series. From this ensemble, we calculate the biomass trajectories and then compute the posterior estimate of (11) as the proportion of biomass trajectories that exceed $B_{MSY}$ after 10 years.

**Data availability.** All data analysed in this study are publicly available for download at www.ramlegacy.org.

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

## Acknowledgements

We are indebted to all the stock assessment scientists whose dedicated work has made these findings possible. We also gratefully acknowledge the Natural Sciences and Engineering Research Council of Canada and the Sobey Fund for Oceans for support. We thank J. Hutchings, J. Mills-Flemming, S. Ambrose and C. Walters for discussions, and D. Hively and R. Hilborn for data.

## Author contributions

G.L.B., M.D. and B.W. designed the study; G.L.B., M.D. and L.K. developed the model; G.L.B. performed the analysis; G.L.B., M.D., L.K. and B.W. wrote the paper.

## Additional information

**Competing interests:** The authors declare no competing fianancial interests.

**Publisher's note**: 

