## [Peer Review File · Nature Communications]

Reviewers' Comments:

Reviewer #1 (Remarks to the Author):

This paper is a potentially interesting twist on vert-Pre et al. 2013. Britten et al. use the same dataset to investigate the possibility of both trends and periodic shifts in productivity of exploited marine fish populations. However, the results here are almost pre-determined by the way the model is set up. This is not necessarily a fatal flaw, but I think it does require comparing this model to others which are just as reasonable.

For example, the "finding" of "persistent (autocorrelated) and periodic shifts relative to the historical productivity stage" is all but guaranteed from the random walk with drift error structure (Eq. 2). The intrinsic growth rate is autocorrelated (i.e., correlated with its value in the previous time step) by definition because that is how the model is set up. The model could just as logically have been constructed using a non-stationary intrinsic growth rate that was allowed to vary around a mean value which may have a trend. That is:

$$r_{t+1} = \mu_r + \delta * t + \text{error}$$

where μ_r is the initial mean value of r in year 0, δ is the annual change in r , and error is the same normally distributed error in Eq. 2. A real test for "persistent (autocorrelated) and periodic shifts" would involve analysis of the error term from this model. If there is real autocorrelation in productivity, these estimated annual deviations should be autocorrelated.

Similarly, much is made of the fact that there was "a substantial non-zero probability density for the productivity standard deviation" (line 81-82). The way the mode is set up, any lack of fit will result in a non-zero probability density for σ_r . Essentially, all this means is that a basic Schaefer model doesn't fit perfectly and if you allow r to vary, it will fit better. This doesn't say anything substantial about r . If you allowed K to vary, it would also fit better. If you allowed annual deviations in surplus production, it would also fit better. For any stock, there is no unique decomposition of the variance around the Schaefer model predictions. If you choose to model this variance as a random walk for the r parameter, then you shouldn't be surprised when that's where the variability shows up.

My recommendation is to reframe this analysis as a model selection problem. The null model explains changes in surplus production (SP) using the Schaefer model with fixed r and K . Are the deviations between observed and predicted SP autocorrelated and do they indicate the existence of productivity regimes? The four alternative models would then be two models in which K or r are allowed to vary according to eq. 2 and two models in which K or r are allowed to vary according to the error model above (random deviations around a mean with (possible) trend). How well do these different models predict the data? Do the errors themselves show autocorrelation?

If the main results hold up after this more thorough analysis, I think this paper will be novel and interesting enough to publish in Nature Communications. As it stands now, though, the way the problem is set up determines the answer.

A less critical, but still important problem with the ms in its current state is that it sets up a bit of a straw man in discussions of what is currently known or what fishery managers and stock assessment scientists currently do. There is a large and growing literature on non-stationarity in stock assessment models. On the academic side, there are many examples of models that allow for changes in natural mortality or fishery selectivity. On the operational assessment side, the most widely used stock assessment model (Stock Synthesis) allows the option for recruitment to vary as a random walk, much as the intrinsic rate of growth is allowed to vary in Britten et al.'s analysis. On the management side, projections of rebuilding timelines routinely address changes in recruitment (the dominant component of changes in productivity) by drawing future recruitments from a distribution centered on the mean of recent values rather than the whole timeseries. The manuscript would benefit from a more accurate acknowledgment of the current state of affairs with respect to addressing non-stationarity in fish population dynamics.

Specific comments

line 47-48 - "Previous analyses... have assumed static productivity." This is not quite accurate. The two analyses mentioned assumed a consistent relationship between productivity and biomass, but productivity in these analyses was not static.

line 216-218 - None of the stock assessment model outputs in the RAM Legacy database can honestly be described as "empirically derived and not generated as output from a deterministic fisheries management model." All of the biomass time series are model outputs. Those models permit various amounts of process error and thus generate time series with different amounts of variability in biomass, but none of them are raw observations of biomass. This criteria needs to be described more specifically. Which types of assessment models were excluded based on this criterion?

line 2243-245 - This section is awkwardly worded. The phrase "nonstationary in their uncertainty levels" seems to suggest that the error variance is changing through time. I don't think that's what is meant here, but if it is, it should be made explicit.

Reviewer #2 (Remarks to the Author):

The issue of fish stock rebuilding is an important global challenge. Fisheries management has improved greatly in the last several decades, especially in industrialized countries. However, natural variability and increasingly strong trends from climate change present a challenge to traditional management approaches that presume a steady-state. In that vein, this is a potentially important and influential paper. However, I think there are some conceptual challenges that the authors need to address before it is ready to be published, especially in a high profile journal like Nature Communications.

Overall, I think this is a good example of a meta-analytic approach. By looking at a large collection of stocks, this study has the potential to identify large-scale, common patterns. The main goal of the analysis is to understand how accounting for non-stationarity in population growth could impact rebuilding times. While focusing on productivity (r) is necessary, I don't think it is sufficient to really understand the problem. Rebuilding involves setting a harvest policy that allows stock biomass to increase to a desired reference level in a specified time frame. Stock productivity plays an important role in setting the rebuilding rate. If the environment changes such that recruitment is reduced or mortality increases, then the rebuilding time will increase. However, this simple calculation assumes that the reference point (in this case B_{msy}) is unchanged. In reality, a change in productivity will almost certainly involve a change in the carrying capacity of the environment. I would expect that in many cases, reduced productivity would go hand in hand with a reduced carrying capacity. Thus, although we may rebuild more slowly, we also don't have as far to go to rebuild. I think this study would be much more interesting if the authors were to include carrying capacity in the analysis. First of all, just understanding whether r and K co-vary would be noteworthy. Then, including non-stationarity in K would make it possible to more realistically look at rebuilding times.

I understand that shifting reference points is controversial in the conservation community. Many view this as letting fishermen off the hook; however, I suspect that harvest levels would not differ very much (since F_{msy} is proportional to r). More importantly, setting reference points correctly is necessary to make fisheries management accurately reflect conditions in the ocean.

In addition to my general concern about including carrying capacity, I have a few specific comments that I think would strengthen the paper:

p1. title. I wonder about using a technical word like “nonstationary” in the title. What about “changing” or “variable?”

p1. abstract: The phrase “reveals nonstationary stochastic fluctuations in annual productivity that, when accounted for, extends rebuilding timelines for stocks currently depleted below target biomass.” is awkward. The way this is phrased, it seems like a choice—if we don’t account for non-stationarity, we’ll rebuild them faster. I would flip this: assuming stationarity leads to unrealistic, overly optimistic timelines.”

p2, L29. The paper asserts repeatedly that the US MSA mandates 10 year rebuilding timelines. This is not strictly true. Rebuilding timelines can be longer for species that are slow growing or long lived (i.e. rockfish, sharks). Please double check the accuracy of this statement. Frankly, I think it is just a heuristic that has risen to the level of myth.

p2, L36. I would change this from contemporary to contemporary and future.

p2, L41. I would argue that assuming stationarity was a bad idea, even before climate change kicked in. There are lots of examples of non-stationarity in marine ecosystems associated with things like the PDO and NAO and regime shifts caused by both physical processes and overfishing. It was a bad idea that is now even worse, and it is an especially bad assumption for depleted stocks that are the focus of this paper.

p2, L46. You highlight environmental uncertainty. I would also add trends to bring in a climate angle.

p2, L60. Yes, r is fundamental, but it is also entirely theoretical. It merges many fundamental biological processes like reproduction, mortality, and somatic growth.

p4, L99. This paragraph talks in general about stocks, but results are only presented for two stocks. A skeptical reader will wonder whether you are cherry-picking examples that show the strongest contrast. You need to present something that captures the range of relationships in your analysis and then justify why you selected these stocks. To really avoid the cherry-picking allegation, you should pick stocks that contrast—one that shows little effect and one that shows a large effect.

p4, L102. I’m not a fan of the color choices in this paper. There are lots of colorblind-safe color schemes out there, please use one. Incidentally, this should be part of Nature’s style policy.

p4, L113. Don’t capitalize bluefin.

p4, L131. For fisheries management, this is perhaps the most important result. Showing that uncertainty is not symmetric but has a long-tail suggests a higher degree of risk than currently assumed. Just knowing that the tails are long should lead to a more precautionary approach.

p4, L137. This is not quite the experiment I would've done. First, it's obvious that increasing F will lead to slower rebuilding. This would be accounted for in standard fishery management, and you'd expect that the management system would find an F that was low enough to get rebuilding under the assumptions of their model. I think it would be more interesting and useful to contrast F assuming stationarity with F using your non-stationary models. Basically, if you assume stationarity, do you pick an F that is wildly off the mark?

Reviewer #3 (Remarks to the Author):

The authors simulate the consequences of a non-static values of r (intrinsic productivity) on rebuilding of fisheries stocks.

The manuscript is well-written and very well illustrated.

Their main findings are:

- there is no obvious trend in r in assessed global stocks
- that assuming variation in r increases the uncertainty surrounding rebuilding times and the subsequent time for rebuilding
- major stocks have had changes to productivity though because r integrates growth and mortality (themselves dependent on many things) causation and prediction are difficult
- least productive stocks are most depleted

Most of these findings are not helpful (though the authors suggest that they will assist fisheries management) nor strictly speaking novel.

The authors suggest that if a bayesian approach to incorporating changes in r is used then fisheries management could avoid some of the past disasters blamed on undetected changes to productivity - even assuming some autocorrelation or stanzas, it would seem that the shifts would not be predictable or detectable in time to make dynamic changes to fisheries management (which have many other sources of inertia anyway) - if not then the authors should argue the case for this better.

Similarly many would assume, without firm evidence to the contrary, that random changes to r around a mean value that is known would not devalue the value of management using the longer

term average value (which allowance for uncertainty) as is currently done. All 'static' parameters describing populations are variable and current management practices (SME etc) incorporate this fact.

How much of the change in r is fishing induced?

What is meant by the ref on line 60?

POINT BY POINT AUTHOR RESPONSE:

Reviewer #1 (Remarks to the Author):

REVIEWER: This paper is a potentially interesting twist on vert-Pre et al. 2013. Britten et al. use the same dataset to investigate the possibility of both trends and periodic shifts in productivity of exploited marine fish populations. However, the results here are almost pre-determined by the way the model is set up. This is not necessarily a fatal flaw, but I think it does require comparing this model to others which are just as reasonable.

For example, the "finding" of "persistent (autocorrelated) and periodic shifts relative to the historical productivity stage" is all but guaranteed from the random walk with drift error structure (Eq. 2). The intrinsic growth rate is autocorrelated (i.e., correlated with its value in the previous time step) by definition because that is how the model is set up. The model could just as logically have been constructed using a non-stationary intrinsic growth rate that was allowed to vary around a mean value which may have a trend. That is:

$$r_{t+1} = \mu_r + \delta * t + \text{error}$$

where μ_r is the initial mean value of r in year 0, δ is the annual change in r , and error is the same normally distributed error in Eq. 2. A real test for "persistent (autocorrelated) and periodic shifts" would involve analysis of the error term from this model. If there is real autocorrelation in productivity, these estimated annual deviations should be autocorrelated.

AUTHOR RESPONSE: The reviewer makes the excellent point that the added flexibility provided by the random walk dynamics will inevitably capture some degree of additional variation in the data. It is therefore important to justify the time-varying dynamics on a stock-by-stock basis. **In light of this point, we now perform an autocorrelation analysis of raw biomass deviations as the reviewer suggests** (described on lines 73-82, 119-124, 275-280). In particular, we fit a static Graham-Schaefer model to each individual time series to extract the biomass residuals and fit a first-order autoregressive model. A stock was characterized as nonstationary if the 95% credible interval for the autoregressive parameter excludes zero. We then only fit the time-varying Graham-Schaefer model to those stocks classified as nonstationary.

REVIEWER: Similarly, much is made of the fact that there was "a substantial non-zero probability density for the productivity standard deviation" (line 81-82). The way the mode is set up, any lack of fit will result in a non-zero probability density for σ_r . Essentially, all this means is that a basic Schaefer model doesn't fit perfectly and if you allow r to vary, it will fit better. This doesn't say anything substantial about r . If you allowed K to vary, it would also fit better. If you allowed annual deviations in surplus production, it would also fit better. For any stock, there is no unique decomposition of the variance around the Schaefer model predictions. If you choose to model this variance as a random walk for the r parameter, then you shouldn't be surprised when that's where the variability shows up.

AUTHOR RESPONSE: This point is related to the previous comment regarding added flexibility of the time-varying models. While we agree that the additional degrees of freedom afforded by the time-varying parameters will inevitably pick up some additional variation, we disagree that the time-varying signal is predetermined simply by a poor static fit. The reason is that the state space model effectively performs a variance partitioning with respect to the residual variation. Lack of fit is either captured by uncorrelated 'measurement error' or persistent shifts in the underlying productivity state. So in the context of our models, it is more accurate that any *autocorrelated* lack of fit will result in non-zero probability density for σ_r , whereas uncorrelated lack of fit will tend to show up in the measurement error. **The additional point regarding whether to capture the autocorrelated signal via time-varying r vs. time-varying K has become a major component of our revisions, the details of which are described below.**

REVIEWER: My recommendation is to reframe this analysis as a model selection problem. The null model explains changes in surplus production (SP) using the Schaefer model with fixed r and K . Are the deviations between observed and predicted SP autocorrelated and do they indicate the existence of productivity regimes? The four alternative models would then be two models in which K or r are allowed to vary according to eq. 2 and two models in which K or r are allowed to vary according to the error model above (random deviations around a mean with (possible) trend). How well do these different models predict the data? Do the errors themselves show autocorrelation?

If the main results hold up after this more thorough analysis, I think this paper will be novel and interesting enough to publish in Nature Communications. As it stands now, though, the way the

problem is set up determines the answer.

AUTHOR RESPONSE: We have **fully implemented the reviewer's excellent suggestion to frame the nonstationary analysis as a model selection problem** by only analyzing the stocks that display persistent autocorrelated behaviour in their biomass residuals. In accordance with the reviewer's suggestion to investigate variation in K , **we have implemented a full simulation analysis with respect to the effect of variation in K on biomass forecasts**. Specifically, we simulated time-varying K under the assumption of a stochastic random walk with variance set at 5, 10, and 20% of mean posterior K . In general we found that variation in K reduced recovery time probabilities (making the impacts of stochastic parameters more severe), while the general pattern in terms of the relative impacts of fishing remained intact. The main reason we did not include formal estimation of K is that initial simulation studies revealed that known changes in time-varying K to be much more difficult to re-estimate relative to r using the Bayesian algorithms. This is due to the weak numerical impact of K on biomass predictions when stocks become depleted. Estimating simultaneous change in r and K is also not statistically possible due to an inherent non-identifiability problem. For these reasons we focused on formal estimation of r and present time-varying K as a simulation study with respect to biomass forecasting. **These new methods and results are now fully implemented in the revision** (lines 108-116, 166-170, 227-240, Figures S1, S2) **and have strengthened our overall conclusions**. We thank the reviewer for his/her thoughtful suggestions.

REVIEWER: A less critical, but still important problem with the ms in its current state is that it sets up a bit of a straw man in discussions of what is currently known or what fishery managers and stock assessment scientists currently do. There is a large and growing literature on non-stationarity in stock assessment models. On the academic side, there are many examples of models that allow for changes in natural mortality or fishery selectivity. On the operational assessment side, the most widely used stock assessment model (Stock Synthesis) allows the option for recruitment to vary as a random walk, much as the intrinsic rate of growth is allowed to vary in Britten et al.'s analysis. On the management side, projections of rebuilding timelines routinely address changes in recruitment (the dominant component of changes in productivity) by drawing future recruitments from a distribution centered on the mean of recent values rather than the whole timeseries. The manuscript would benefit from a more accurate acknowledgment of the current state of affairs with respect to addressing non-stationarity in fish population dynamics.

AUTHOR RESPONSE: We agree and have now tempered our claims of novelty with respect to modeling nonstationary parameters. Rather, we highlight the novelty of our analysis in terms of the analysis of hundreds of globally distributed stocks and the general implications of stochastic parameter for biomass forecasts of depleted stock and associated rebuilding timelines; we also specifically highlight Safina et al. 2005 as a more direct contrast with respect to static vs. stochastic recovery time analysis (lines 51-60, 185-189).

REVIEWER:

Specific comments

line 47-48 - "Previous analyses... have assumed static productivity." This is not quite accurate.

The two analyses mentioned assumed a consistent relationship between productivity and biomass, but productivity in these analyses was not static.

AUTHOR RESPONSE: This is a mistake on our part for not being clear in our terminology. We meant ‘productivity’ to be equal to the productivity parameter r and not annual biomass production $rB(1-B/K)$ which has been assumed time-varying previously (e.g. Vert-pre et al. 2013). **We now specifically refer to r which was assumed static in cited global analyses of recovery time** (lines 191-195).

REVIEWER: line 216-218 - None of the stock assessment model outputs in the RAM Legacy database can honestly be described as "empirically derived and not generated as output from a deterministic fisheries management model." All of the biomass time series are model outputs. Those models permit various amounts of process error and thus generate time series with different amounts of variability in biomass, but none of them are raw observations of biomass. This criteria needs to be described more specifically. Which types of assessment models were excluded based on this criterion?

AUTHOR RESPONSE: The reviewer makes an excellent point that the time series used in this study originate from very different stock assessment methods which result in biomass time series of different statistical character. This is an issue we’ve given considerable thought to in the past. As there is no ‘algorithm’ to properly select data amongst such heterogeneous data, we had previously relied on qualitative judgement based on reviewing the individual stock assessments. **To avoid this potential ambiguity, we have now relaxed this and have included all stocks for which there is a direct estimate of B and C. We have, however, removed retrospective analyses from individual time series via visual inspection.** Retrospective analyses are identifiable where the time series become smooth and does not contain stochastic variation from year to year. This relaxed criteria added 21 stocks to the analysis. We describe the new selection criteria on lines 263-270.

REVIEWER: line 2243-245 - This section is awkwardly worded. The phrase "nonstationary in their uncertainty levels" seems to suggest that the error variance is changing through time. I don't think that's what is meant here, but if it is, it should be made explicit.

AUTHOR RESPONSE: We have removed this wording. We thank the reviewer for his/her time in providing constructive comments to the manuscript.

Reviewer #2 (Remarks to the Author):

REVIEWER: The issue of fish stock rebuilding is an important global challenge. Fisheries management has improved greatly in the last several decades, especially in industrialized countries. However, natural variability and increasingly strong trends from climate change present a challenge to traditional management approaches that presume a steady-state. In that vein, this is a potentially important and influential paper. However, I think there are some conceptual challenges that the authors need to address before it is ready to be published, especially in a high profile journal like Nature Communications.

Overall, I think this is a good example of a meta-analytic approach. By looking at a large collection of stocks, this study has the potential to identify large-scale, common patterns. The main goal of the analysis is to understand how accounting for non-stationarity in population growth could impact rebuilding times. While focusing on productivity (r) is necessary, I don't think it is sufficient to really understand the problem. Rebuilding involves setting a harvest policy that allows stock biomass to increase to a desired reference level in a specified time frame. Stock productivity plays an important role in setting the rebuilding rate. If the environment changes such that recruitment is reduced or mortality increases, then the rebuilding time will increase. However, this simple calculation assumes that the reference point (in this case B_{msy}) is unchanged. In reality, a change in productivity will almost certainly involve a change in the carrying capacity of the environment. I would expect that in many cases, reduced productivity would go hand in hand with a reduced carrying capacity. Thus, although we may rebuild more slowly, we also don't have as far to go to rebuild. I think this study would be much more interesting if the authors were to include carrying capacity in the analysis. First of all, just understanding whether r and K co-vary would be noteworthy. Then, including non-stationarity in K would make it possible to more realistically look at rebuilding times. I understand that shifting reference points is controversial in the conservation community. Many view this as letting fishermen off the hook; however, I suspect that harvest levels would not differ very much (since F_{msy} is proportional to r). More importantly, setting reference points correctly is necessary to make fisheries management accurately reflect conditions in the ocean.

AUTHOR RESPONSE: We thank the reviewer for his/her excellent comments. As outlined in our response to reviewer #1, **we have revised the manuscript by incorporating uncertainty and possible stochasticity of the carrying capacity K in several ways.** First, we now estimate it as an unknown parameter (instead of fixing it at the maximum of annual biomass plus catch) therefore incorporating its uncertainty into all subsequent calculations. Secondly, we have performed a full sensitivity analysis to characterize the consequences of time-varying K for biomass recovery forecasts.

**Comments pasted from AUTHOR RESPONSE to reviewer #1*

Specifically, we simulated time-varying K under the assumption of a stochastic random walk with variance set at 5, 10, and 20% of mean posterior K . In general we found that variation in K reduced recovery time probabilities (making the impacts of stochastic parameters more severe), while the general pattern in terms of the relative impacts of fishing remained intact. The main reason we did not include formal estimation of K is that initial simulation studies revealed that known changes in time-varying K to be much more difficult to re-estimate, relative to r , using

the Bayesian algorithms. This is due to the weak numerical impact of K on biomass predictions when stocks become depleted. Estimating simultaneous change in r and K is also not statistically possible due to an inherent non-identifiability problem. For these reasons we focused on formal estimation of r and present time-varying K as a simulation study with respect to biomass forecasting. **These new methods and results are now fully implemented in the revision** (lines 108-116, 166-170, 227-240, Figures S1, S2) **and have strengthened our overall conclusions.** We thank the reviewer for his/her thoughtful suggestions.

REVIEWER: In addition to my general concern about including carrying capacity, I have a few specific comments that I think would strengthen the paper:

p1. title. I wonder about using a technical word like “nonstationary” in the title. What about “changing” or “variable”?

AUTHOR RESPONSE: Agreed. While we like the term, we agree that it is technical and may detract from the title geared for a general audience. The title has been revised to the reviewer’s suggestion of “changing”.

REVIEWER: p1. abstract: The phrase “reveals nonstationary stochastic fluctuations in annual productivity that, when accounted for, extends rebuilding timelines for stocks currently depleted below target biomass.” is awkward. The way this is phrased, it seems like a choice—if we don’t account for non-stationarity, we’ll rebuild them faster. I would flip this: assuming stationarity leads to unrealistic, overly optimistic timelines.”

AUTHOR RESPONSE: Agreed. The wording has been changed to reflect that recovery times are forecast from static productivity *models*.

REVIEWER: p2, L29. The paper asserts repeatedly that the US MSA mandates 10 year rebuilding timelines. This is not strictly true. Rebuilding timelines can be longer for species that are slow growing or long lived (i.e. rockfish, sharks). Please double check the accuracy of this statement. Frankly, I think it is just a heuristic that has risen to the level of myth.

AUTHOR RESPONSE: Agreed. This is an important point. The ten year rebuilding time is indeed only a guide post in the MSA while other factors of the specific biological characteristics of the stock are taken into account in setting the actual mandate for any particular stock. We have removed any suggestion of a strict ten year recovery plan and now present the choice of a ten year as a default timescale, while noting that our general conclusions apply equally to any recovery timescale (lines 103-106).

REVIEWER: p2, L36. I would change this from contemporary to contemporary and future.

AUTHOR RESPONSE: Agreed. This wording change has been made (lines 36-40).

REVIEWER: p2, L41. I would argue that assuming stationarity was a bad idea, even before climate change kicked in. There are lots of examples of non-stationarity in marine ecosystems associated with things like the PDO and NAO and regime shifts caused by both physical

processes and overfishing. It was a bad idea that is now even worse, and it is an especially bad assumption for depleted stocks that are the focus of this paper.

AUTHOR RESPONSE: We completely agree. We have added a statement pointing to the fact that stationary was likely a poor a priori assumption, but can no longer be ignored based on several recent empirical analyses (lines 45-49).

REVIEWER: p2, L46. You highlight environmental uncertainty. I would also add trends to bring in a climate angle.

AUTHOR RESPONSE: Agreed. We have corrected the phrase to refer to ‘climate and environmental uncertainty’ (line 45-49).

REVIEWER: p2, L60. Yes, r is fundamental, but it is also entirely theoretical. It merges many fundamental biological processes like reproduction, mortality, and somatic growth.

AUTHOR RESPONSE: We agree that r is a ‘net’ rate. But in that sense it is empirical and can be diagnosed from observations of biomass alone. We make the point in the MS that r combines these processes (lines 189-205).

REVIEWER: p4, L99. This paragraph talks in general about stocks, but results are only presented for two stocks. A skeptical reader will wonder whether you are cherry-picking examples that show the strongest contrast. You need to present something that captures the range of relationships in your analysis and then justify why you selected these stocks. To really avoid the cherry-picking allegation, you should pick stocks that contrast—one that shows little effect and one that shows a large effect.

AUTHOR RESPONSE: Agreed. These stocks are chosen to be large and recognizable stocks with easily interpretable patterns of productivity variation. Based on our model selection procedure described the response to another reviewer we now exclude stocks from the analysis that do not exhibit appreciably stochastic variation in their productivity state. So the magnitudes of stochastic productivity variation displayed in Fig. 2 are now representative of all stocks used in the analysis. The detailed of the model selection are described on lines 73-82, 119-124, 275-280 of the revised MS.

REVIEWER: p4, L102. I’m not a fan of the color choices in this paper. There are lots of colorblind-safe color schemes out there, please use one. Incidentally, this should be part of Nature’s style policy.

AUTHOR RESPONSE: Agreed. We have removed Fig. 1 which used weakly contrasted colors. The only colors used in the MS are now black, grey, red, and green.

REVIEWER: p4, L113. Don’t capitalize bluefin.

AUTHOR RESPONSE: Agreed and changed.

REVIEWER: p4, L131. For fisheries management, this is perhaps the most important result. Showing that uncertainty is not symmetric but has a long-tail suggests a higher degree of risk than currently assumed. Just knowing that the tails are long should lead to a more precautionary approach.

AUTHOR RESPONSE: We completely agree. Because the revision now highlights the results using a different figure we no longer display the heavy tail distribution, but rather point it out in text and describe why it is a poor characterization of recovery (lines 310-312). The new results regarding the additional consequences of stochastic K also strengthens this point.

REVIEWER: p4, L137. This is not quite the experiment I would've done. First, it's obvious that increasing F will lead to slower rebuilding. This would be accounted for in standard fishery management, and you'd expect that the management system would find an F that was low enough to get rebuilding under the assumptions of their model. I think it would be more interesting and useful to contrast F assuming stationarity with F using your non-stationary models. Basically, if you assume stationarity, do you pick an F that is wildly off the mark?

AUTHOR RESPONSE: The reviewer makes an excellent point regarding the effect of static versus time-varying fishing mortality informed by time-varying productivity. We have investigated these dynamics in simulation and have found some very interesting results. In general, the mean biomass forecast was not found to be appreciably different under the two regimes, whereas the biomass variance is much higher under the static fishing regime. We believe these results would require more in depth analysis to be reported and interpreted in full, so we therefore mention this investigation as a sensitivity analysis and avenue of detailed future research (lines 175-182). We thank the reviewer for all the thoughtful suggestions which we believe have improved the MS.

Reviewer #3 (Remarks to the Author):

REVIEWER: The authors simulate the consequences of a non-static values of r (intrinsic productivity) on rebuilding of fisheries stocks.

AUTHOR RESPONSE: To clarify, there are two major components to the paper. One is the analysis of 276 time series to empirically characterize stochastic productivity; the second is the simulation of consequences of stochastic parameters using variance parameters estimated from the data.

REVIEWER: The manuscript is well-written and very well illustrated.

Their main findings are:

- there is no obvious trend in r in assessed global stocks
- that assuming variation in r increases the uncertainty surrounding rebuilding times and the subsequent time for rebuilding
- major stocks have had changes to productivity though because r integrates growth and mortality (themselves dependent on many things) causation and prediction are difficult

-least productive stocks are most depleted

Most of these findings are not helpful (though the authors suggest that they will assist fisheries management) nor strictly speaking novel.

AUTHOR RESPONSE: As far as we know, a global meta-analysis of stochastic productivity variation in 276 globally distributed stocks and quantitative modeling of the consequences for recovery time has not been performed previously. **We have carefully revised the text to outline the probable consequences of our findings for fisheries management and marine ecology more broadly.**

The authors suggest that if a bayesian approach to incorporating changes in r is used then fisheries management could avoid some of the past disasters blamed on undetected changes to productivity - even assuming some autocorrelation or stanzas, it would seem that the shifts would not be predictable or detectable in time to make dynamic changes to fisheries management (which have many other sources of inertia anyway) - if not then the authors should argue the case for this better.

Similarly many would assume, without firm evidence to the contrary, that random changes to r around a mean value that is known would not devalue the value of management using the longer term average value (which allowance for uncertainty) as is currently done. All 'static' parameters describing populations are variable and current management practices (SME etc) incorporate this fact.

How much of the change in r is fishing induced?

AUTHOR RESPONSE: While the reviewer raises a very interesting question regarding the role of fishing in altering the intrinsic productivity of stocks, we note that determining the causal factors driving changes in the underlying productivity state is outside the scope of this paper. From the biomass time series provided in the RAM Legacy database, we can only recover the net rate of change and cannot further decompose r into its constituent biological processes – e.g. somatic growth, age-distribution - some of which would be effected by harvesting (and also the history of harvesting). **We highlight in the manuscript that studying the individual processes causing these changes is an important avenue for follow up research** (lines 198-205).

What is meant by the ref on line 60?

AUTHOR RESPONSE: Our mistake. We removed the `_ref.` part which and just display the ref. number as per Nature style. The reference is to a paper where a late and highly prominent fisheries scientist referred to the parameter r (the focus of this study) as the `_the most fundamental of all ecological parameters.` We believe this reference provides qualitative context for the general reader unfamiliar with the field of population dynamics.

Reviewers' Comments:

Reviewer #1 (Remarks to the Author):

The authors have done an unusually thorough job of addressing the reviewer comments (both my own and those of the other reviewers). In particular, the addition of an initial test for autocorrelation in productivity and the simulation analysis of changes in carrying capacity (K) sufficiently address my most substantial comments.

I think the revised manuscript will make an important contribution to the growing literature on managing fisheries in a changing climate.

Reviewer #2 (Remarks to the Author):

My main critique of the original draft to this paper focused on the need to consider both carrying capacity (K) alongside intrinsic growth (r). I think it is notable that the reviewers converged on this issue. I am pleased that the authors took this advice seriously, and I think the carrying capacity work makes the paper much stronger. However, the analysis isn't quite the way I was thinking of it. I liked how one of the other reviewers framed this as a model selection problem. Essentially, for each non-stationary stock (and I really liked the stationarity criterion), this would mean fitting three models: one with varying r and fixed K, one with fixed r and varying K, and one with both varying, and then choosing among the three. As currently formulated, K is viewed as being secondarily important. Perhaps there is a technical reason why this makes sense?

With that said, I think the paper is much stronger and getting much closer to what I would want to see in Nature Communications. I think the conclusions about recovery times and risk in management are much stronger, and I think it is an important contribution. In general, I would have appreciated more discussion on the ecological or fisheries implications and less space devoted to the modeling techniques. While the Graham-Schafer model with the non-stationary terms proved to be a useful tool for pulling patterns out of the RAM database, the models themselves are not the point.

pg 1, L35-36. The sentence starting "We hereafter.." is not clear to me. Perhaps "Stocks that increase to Bmsy from a depressed state are considered recovered."

pg 2, L49. Change "whether" to "how"

pg 2, L74-76. This sentence is awkward and I think it may be inverted. Isn't it the opposite—that

is, if zero is not in the interval, then there is a significant autoregressive term?

pg 3, L107-108. Unclear to me why variance in K is considered secondary to variance in r. They are both going to vary, likely together. I would think estimating both simultaneously (in a non-hierarchical way) would be the most ecological and statistically justifiable treatment. As the other reviewer suggested, this is essentially a model-selection problem: for non-stationary models, does variance in r, K, or both offer the best explanation for variability in the time series?

pg 4, L131. Unless this species is named after Prof. Bluefin, then it shouldn't be capitalized.

pg 4, L139. Contrasting red and green is the opposite of colorblind-safe.

pg 6, L198-201. This sentence doesn't make sense to me. I don't quite see the connection between this study (about changes in r and K) and top-down effects. Are you talking about some kind of depensation?

Reviewer #3 (Remarks to the Author):

On balance this should be an important paper and is improved.

Fusion of findings within commonly used management frameworks like MSE would facilitate uptake and I encourage it.

line 75 is confusing.

Reviewer #1:

The authors have done an unusually thorough job of addressing the reviewer comments (both my own and those of the other reviewers). In particular, the addition of an initial test for autocorrelation in productivity and the simulation analysis of changes in carrying capacity (K) sufficiently address my most substantial comments.

I think the revised manuscript will make an important contribution to the growing literature on managing fisheries in a changing climate.

RESPONSE: We thank the reviewer for his/her support for the manuscript and effort in strengthening the results and conclusions.

Reviewer #2:

My main critique of the original draft to this paper focused on the need to consider both carrying capacity (K) alongside intrinsic growth (r). I think it is notable that the reviewers converged on this issue. I am pleased that the authors took this advice seriously, and I think the carrying capacity work makes the paper much stronger.

RESPONSE: We thank the reviewer for his/her constructive comments.

However, the analysis isn't quite the way I was thinking of it. I liked how one of the other reviewers framed this as a model selection problem. Essentially, for each non-stationary stock (and I really liked the stationarity criterion), this would mean fitting three models: one with varying r and fixed K, one with fixed r and varying K, and one with both varying, and then choosing among the three. As currently formulated, K is viewed as being secondarily important. Perhaps there is a technical reason why this makes sense?

RESPONSE: We agree with the reviewer that the ideal setup would test the three outlined models, and we did run extensive simulations to test this approach. Unfortunately, the model selection problem with respect to time-varying K proved unreliable in these simulation and parameter recovery studies. We have expanded the discussion of this point on lines 114-120. On those lines we write: "This difficulty arises due to the weak dynamical connection between biomass and K_t when biomass is low. In contrast, r_t is directly proportional to the stock rate of change across a wide range of biomass levels, yielding more reliable estimation when averaging over possible biomass states. For these reasons we focused the formal time series estimation on r_t and present K_t modeling results in sections of the paper concerning its effects on biomass forecasting and recovery timelines."

With that said, I think the paper is much stronger and getting much closer to what I would want

to see in Nature Communications. I think the conclusions about recovery times and risk in management are much stronger, and I think it is an important contribution. In general, I would have appreciated more discussion on the ecological or fisheries implications and less space devoted to the modeling techniques. While the Graham-Schafer model with the non-stationary terms proved to be a useful tool for pulling patterns out of the RAM database, the models themselves are not the point.

RESPONSE: We thank the reviewer for his/her support of the manuscript. In accordance with the comment, we have now expanded the discussion regarding the fisheries ecology and management implications of time-varying productivity. Specifically we included new discussion on the increasing need for adaptive management when the environment is experiencing unusual rates of change (lines 213-222). We also discuss the importance in understanding the relative time-lag between changes in recruitment and total productivity (206-211), management implications of time-varying carrying capacity (242-254), and the potential socioeconomic implications for missing recovery timelines (256-268).

pg 1, L35-36. The sentence starting "We hereafter.." is not clear to me. Perhaps "Stocks that increase to B_{MSY} from a depressed state are considered recovered."

RESPONSE: We have edited the sentence to read: "We hereafter use the term *recovery* to refer to the event when a depleted stock grows to exceed B_{MSY} "

pg 2, L49. Change "whether" to "how"

RESPONSE: Changed. Thank you.

pg 2, L74-76. This sentence is awkward and I think it may be inverted. Isn't it the opposite—that is, if zero is not in the interval, then there is a significant autoregressive term?

RESPONSE: We thank the reviewer for catching this most unfortunate typo! Indeed there was a 'not' missing from the sentence and has been changed.

pg 3, L107-108. Unclear to me why variance in K is considered secondary to variance in r . They are both going to vary, likely together. I would think estimating both simultaneously (in a non-hierarchical way) would be the most ecological and statistically justifiable treatment. As the other reviewer suggested, this is essentially a model-selection problem: for non-stationary models, does variance in r , K , or both offer the best explanation for variability in the time series?

RESPONSE: We point the reviewer to a previous response where we detail the difficulties with a formal historical estimation of time-varying K and its variance σ_K^2 , along with the documented parameter-recovery simulation and sensitivity analyses. That said, we believe this additional analysis has important implications for management based on the simulated effects on biomass trajectories. These simulated effects were not included in the original submission and their inclusion has significantly improved the paper. We discuss these management implications on lines 240-254.

pg 4, L131. Unless this species is named after Prof. Bluefin, then it shouldn't be capitalized.

RESPONSE: Regrettably we know of no Prof. Bluefin. We have thus changed the letter to lowercase and appreciate the humor.

pg 4, L139. Contrasting red and green is the opposite of colorblind-safe.

RESPONSE: We have changed the green shading to grey in Figure 2. Thank you for pointing this out.

pg 6, L198-201. This sentence doesn't make sense to me. I don't quite see the connection between this study (about changes in r and K) and top-down effects. Are you talking about some kind of depensation?

RESPONSE: To make this clearer, we have explicated stated mortality via top-down predation as a potential driver of productivity changes (line 203-206). This is because the intrinsic productivity parameter includes all sources of natural mortality, including disease, environmental effects, and top down predation.

Reviewer #3:

On balance this should be an important paper and is improved.

Fusion of findings within commonly used management frameworks like MSE would facilitate uptake and I encourage it.

RESPONSE: We agree with the reviewer and have expanded our discussion on these points. Specifically we included new discussion on the increasing need for adaptive, non-traditional management when the environment is experiencing unusual rates of change (lines 213-222). We also discuss the importance in understanding the relative time-lag between changes in recruitment and total productivity and the impact on management (206-211), along with the interesting management implications of time-varying carrying capacity (242-254).

line 75 is confusing.

RESPONSE: We have edited this sentence to make the details of the residual analysis more clear (lines 74-79). We thank the reviewer for his/her investment in improving the manuscript.

Reviewers' Comments:

Reviewer #2 (Remarks to the Author):

I really like how this manuscript has developed. I appreciate your attentiveness to my comments and those of the other authors--this process has felt surprisingly collaborative. The editors asked me to look at the revisions to the discussion section. I think the new material is great and accurately points out the implications for fisheries management. The two paragraphs (L246-283) that get into the details of the modeling may be a bit too detailed for Nature. The lead in to the discussion actually says many of the things in this paragraph but in a less technical way. If you or the editors are looking to cut some lines, this would be the place to go. Otherwise, great job!